

# DAE Tools: equation-based object-oriented modelling, simulation and optimisation software

Dragan D. Nikolić[1,2]

[1] DAE Tools Project, Belgrade, Serbia
[2] MABE, Faculty of Science & Engineering, University of Limerick, Limerick, Ireland

## ABSTRACT

In this work, DAE Tools modelling, simulation and optimisation software, its programming paradigms and main features are presented. The current approaches to mathematical modelling such as the use of modelling languages and general-purpose programming languages are analysed. The common set of capabilities required by the typical simulation software are discussed, and the shortcomings of the current approaches recognised. A new hybrid approach is introduced, and the modelling languages and the hybrid approach are compared in terms of the grammar, compiler, parser and interpreter requirements, maintainability and portability. The most important characteristics of the new approach are discussed, such as: (1) support for the runtime model generation; (2) support for the runtime simulation set-up; (3) support for complex runtime operating procedures; (4) interoperability with the third party software packages (i.e. NumPy/SciPy); (5) suitability for embedding and use as a web application or software as a service; and (6) code-generation, model exchange and co-simulation capabilities. The benefits of an equation-based approach to modelling, implemented in a fourth generation object-oriented general purpose programming language such as Python are discussed. The architecture and the software implementation details as well as the type of problems that can be solved using DAE Tools software are described. Finally, some applications of the software at different levels of abstraction are presented, and its embedding capabilities and suitability for use as a software as a service is demonstrated.

## INTRODUCTION

In general, two main approaches to mathematical modelling currently exist: (a) use of modelling languages, either domain specific or multi-domain such as Modelica (*Fritzson & Engelson, 1998*), Ascend (*Piela et al., 1991*), gPROMS (*Barton & Pantelides, 1994*), GAMS (*Brook, Kendrick & Meeraus, 1988*), Dymola (*Elmqvist, 1978*), APMonitor (*Hedengren et al., 2014*), and (b) use of general-purpose programming languages, either lower level third-generation languages such as C, C++ and Fortran (i.e. PETSc—a suite of data structures and routines for the scalable solution of scientific applications, *Balay et al., 2015*, and SUNDIALS—Suite of Nonlinear and Differential/Algebraic Equation Solvers, *Hindmarsh et al., 2005*), or higher level fourth-generation languages such as Python (i.e. Assimulo—a high-level interface for a wide variety of ODE/DAE solvers

Corresponding author
Dragan D. Nikolić,
dnikolic@daetools.com

written in C and Fortran, *Andersson, Fuhrer & Akesson, 2015*) and multi-paradigm numerical languages: MATLAB (*MathWorks, Inc., 2015*), Mathematica (*Wolfram Research, Inc., 2015*), Maple (*Waterloo Maple, Inc., 2015*), Scilab (*Scilab Enterprises, 2015*), and GNU Octave (*Eaton et al., 2015*). The lower-level general purpose languages are also often used for the development of the efficient, tailor-made software (i.e. large-scale finite difference and finite element solvers) targeting one of the available high-performance computing architectures such as General Purpose Graphics Processing Units (GPGPU), Field-Programmable Gate Arrays (FPGA), vector processors and Data Flow Engines (DFE). In addition, some modelling tools provide the Python scripting interface to the simulator engine: APMonitor, JModelica (*Akesson et al., 2010*), and OpenModelica (*Fritzson et al., 2005*); however, their API is limited to the loading of developed models, execution of simulations and processing of the results only. Domain Specific Languages (DSL) are special-purpose programming or specification languages dedicated to a particular problem domain and directly support the key concepts necessary to describe the underlying problems. They are created specifically to solve problems in a particular domain and usually not intended to be able to solve problems outside it (although that may be technically possible in some cases). More versatile, multi-domain modelling languages (such as Modelica or gPROMS) are capable of solving problems in different application domains. Despite their versatility, modelling languages commonly lack or have a limited access to the operating system, third-party numerical libraries and other capabilities that characterise full-featured programming languages, scripting or otherwise. In contrast, general-purpose languages are created to solve problems in a wide variety of application domains, do not support concepts from any domain, and have a direct access to the operating system, low-level functions and third-party libraries.

The most important tasks required to solve a typical simulation or optimisation problem include: the model specification, the simulation setup, the simulation execution, the numerical solution of the system of algebraic/differential equations, and the processing of the results. Each task may require a call or a chained sequence of calls to other software libraries, the methods in those libraries must be available to be called with no significant additional pre-processing and must be able to operate on shared/common data structures. All of these require a two-way interoperability between the software and third-party libraries. Also, the model structure is often not fully defined beforehand and a runtime generation of models ("on-the-fly") using the results from other software is required. Frequently, simulations can not be limited to a straightforward, step-wise integration in time but the custom user-defined operating procedures are required, which can be performed only using the fully-featured programming languages. In addition, it is often desired to compare/benchmark the simulation results between different simulators. This requires the code-generation and the model-exchange capabilities to automatically generate the source code for the target language or export the model definition to a specified (often simulator-independent) model specification language. Exposing the functionality of the developed models to another simulator through a predefined standard interface such as the CAPE-OPEN (http://www.colan.org) and Functional Mock-up Interface (FMI, http://www.fmi-standard.org) is another common

functionality. Finally, the current trends in IT industry show that there is a high demand for cloud solutions, such as Software as a Service (SaaS), Platform as a Service (PaaS) and web applications.

A modelling language implemented as a single monolithic software package can rarely deliver all capabilities required. For instance, the Modelica modelling language allows calls to "C" functions from external shared libraries but with some additional pre-processing. Simple operating procedures are supported directly by the language but they must be embedded into a model, rather than separated into an independent section or function. gPROMS also allows very simple operating procedures to be defined as tasks (only in simulation mode), and user-defined output channels for custom processing of the results. The runtime model generation and complex operating procedures are not supported. Invocation from other software is either not possible or requires an additional application layer. On the other hand, Python, MATLAB and the software suites such as PETSc have an access to an immense number of scientific software libraries, support runtime model generation, completely flexible operating procedures and processing of the results. However, the procedural nature and lack of object-oriented features in MATLAB and absence of fundamental modelling concepts in all three types of environments make development of complex models or model hierarchies difficult.

In this work, a new approach has been proposed and implemented in DAE Tools software which offers some of the key advantages of the modelling languages coupled with the power and flexibility of the general-purpose languages. It is a type of hybrid approach–it is implemented using the general-purpose programming languages such as C++ and Python, but provides the Application Programming Interface (API) that resembles a syntax of modelling languages as much as possible and takes advantage of the higher level general purpose languages to offer an access to the operating system, low-level functions and large number of numerical libraries to solve various numerical problems. To illustrate the new concept, the comparison between Modelica and gPROMS grammar and DAE Tools API for a very simple dynamical model is given in the Source Code Listings 1–3, respectively. The model represents a cylindrical tank containing a liquid inside with an inlet and an outlet flow where the outlet flowrate depends on the liquid level in the tank. It can be observed that the DAE Tools API mimics the expressiveness of the grammar of modelling languages to provide the key modelling concepts while retaining the full power of general purpose programming languages. More details about the API is given in the section *Architecture*.

---

**Listing 1 Buffer Tank model (Modelica)**

```
model BufferTank
  // Import libs
  import Modelica.Math.*;

  parameter Real Density;
  parameter Real CrossSectionalArea;
  parameter Real Alpha;
```

```
    Real HoldUp(start = 0.0);
    Real FlowIn;
    Real FlowOut;
    Real Height;

equation
    // Mass balance
    der(HoldUp) = FlowIn − FlowOut;

    // Relation between liquid level and holdup
    HoldUp = CrossSectionalArea * Height * Density;

    // Outlet flowrate as a function of the liquid level
    FlowOut = Alpha * sqrt(Height);
end BufferTank;
```

**Listing 2** Buffer Tank model (gPROMS)

```
PARAMETER
    Density              AS  Real
    CrossSectionalArea   AS  Real
    Alpha                AS  Real

VARIABLE
    HoldUp   AS  Mass
    FlowIn   AS  Flowrate
    FlowOut  AS  Flowrate
    Height   AS  Length

EQUATION
    # Mass balance
    $HoldUp = FlowIn − FlowOut;

    # Relation between liquid level and holdup
    HoldUp = CrossSectionalArea * Height * Density;

    # Outlet flowrate as a function of the liquid level
    FlowOut = Alpha * sqrt(Height);
```

**Listing 3** Buffer Tank model (DAE Tools)

```
class BufferTank(daeModel):
    def __init__(self, Name, Parent = None, Description = ""):
        daeModel.__init__(self, Name, Parent, Description)

        self.Density = daeParameter("Density",  kg/m**3, self)
        self.Area    = daeParameter("Area",     m**2,    self)
        self.Alpha   = daeParameter("Alpha",    unit(),  self)
```

```
    self.HoldUp  = daeVariable("HoldUp",  mass_t,     self)
    self.FlowIn  = daeVariable("FlowIn",  flowrate_t, self)
    self.FlowOut = daeVariable("FlowOut", flowrate_t, self)
    self.Height  = daeVariable("Height",  length_t,   self)

  def DeclareEquations(self):
    # Mass balance
    eq = self.CreateEquation("MassBalance")
    eq.Residual = self.HoldUp.dt() − self.FlowIn() + self.FlowOut()

    # Relation between liquid level and holdup
    eq = self.CreateEquation("LiquidLevelHoldup")
    eq.Residual = self.HoldUp() − self.Area() * self.Height() * self.Density()

    # Outlet flowrate as a function of the liquid level
    eq = self.CreateEquation("OutletFlowrate")
    eq.Residual = self.FlowOut() − self.Alpha() * Sqrt(self.Height())
```

The article is organised in the following way. First, the DAE Tools programming paradigms and the main features are introduced and discussed. Next, its architecture and the software implementation details are analysed. After that, the algorithm for the solution of DAE systems is presented and some basic information on how to develop models in DAE Tools given. Then, two applications of the software are demonstrated: (a) multi-scale modelling of phase-separating electrodes, and (b) a reference implementation simulator for a new domain specific language. Finally, a summary of the most important characteristics of the software is given in the last section.

## MAIN FEATURES AND PROGRAMMING PARADIGMS

DAE Tools is free software released under the GNU General Public Licence. The source code, the installation packages and more information about the software can be found on the website (http://www.daetools.com). Models can be developed in Python or C++, compiled into an independent executable and deployed with no additional run time libraries. Problems that can be solved are initial value problems of implicit form described by a system of linear, non-linear, and partial-differential equations (only index-1 DAE systems, at the moment). Systems modelled can be with lumped or distributed parameters, steady-state or dynamic, and continuous with some elements of event-driven systems such as discontinuous equations, state transition networks and discrete events. Automatic differentiation is supported through the operator overloading technique using the modified ADOL-C library (*Walther & Griewank, 2012*). All DAE Tools libraries are written in standard ANSI/ISO C++. The code is therefore portable across different platforms, and currently runs on all major operating systems such as GNU/Linux, MacOS and Windows. In general, all platforms with the standards compliant C/C++ and Fortran compilers and Boost libraries (http://www.boost.org) are supported. To date, it has been successfully tested on 32/64 bit x86 and ARM architectures making it suitable for use in embedded systems. Object-oriented capabilities allow a hierarchical model decomposition and facilitate the model re-use. A large number of numerical solvers is

supported. Currently, Sundials IDAS (*Hindmarsh et al., 2005*) variable-order, variable-coefficient BDF solver is used to solve DAE systems and calculate sensitivities. IPOPT (*Wächter & Biegler, 2006*), BONMIN (*Bonami et al., 2008*), and NLopt (*Johnson, 2015*) solvers are employed to solve (mixed integer) non-linear programming problems, and a range of direct/iterative and sequential/multi-threaded sparse matrix linear solvers is interfaced such as SuperLU/SuperLU_MT (*Li, 2005*), PARDISO (*Schenk, Wächter & Hagemann, 2007*), Intel PARDISO, and Trilinos Amesos/AztecOO (*Sala, Stanley & Heroux, 2006*).

Broadly speaking, DAE Tools is not a modelling language (such as Modelica and gPROMS) nor an integrated software suite of data structures and routines for scientific applications (such as PETSc), but rather a higher level structure–an architectural design of interdependent software components providing an API for: (a) model development/specification, (b) activities on developed models such as simulation, optimisation and parameter estimation, (c) processing of the results, (d) report generation, and (e) code generation and model exchange. However, it can easily be integrated into a software suite with the Graphical User Interface (GUI), embedded into another software or even run as a web service on the server (as it was demonstrated in the section *NineML domain specific language*). The hybrid approach provides a combination of strengths of both modelling and general purpose programming languages. The most important feature of domain-specific/modelling languages is that they allow solutions to be expressed in the idiom and at the level of abstraction of the problem domain. They directly support all modelling concepts by the language syntax and provide a clean, concise and an elegant way of building model descriptions. Also, modelling languages could be and often are simulator independent making a model exchange easier. However, all of this comes with a price. For instance, the costs of designing, implementing, and maintaining a domain-specific language as well as the tools required to develop with it are high. In all cases, either a compiler or an interpreter with a lexical parser and an Abstract Syntax Tree (AST) must be developed with all burden that comes with it such as processing of the AST, error handling, grammar ambiguities and hidden bugs. In addition, there is a cost of learning a new language versus its limited applicability: users are required to master a new language with yet another language grammar. Integration of modelling languages with other components is difficult and limited by the existence of wrappers around a simulator engine. Models usually cannot be generated in the runtime or at least not easily and cannot be modified in the runtime. Setting up a simulation is specified in the language grammar and it is difficult to do it programmatically. Simulation operating procedures are not fully flexible and manipulation of models is limited to only those operations provided by the language. Finally, the results typically cannot be processed in a user-defined fashion without investing an effort to master the protocol used by the simulator. In contrast, in DAE Tools a compiler/lexical parser/interpreter are an integral part of the programming language (C++ and Python) with a robust error handling, universal grammar and massively tested. No learning of a new language is required, calling external functions/libraries is a built-in feature and models can be created and modified in the runtime. Setting up a simulation is done programmatically and

the initial values can be easily obtained from the other software. Operating procedures are completely flexible (within the limits of a programming language itself) and models can be manipulated in any user-defined way. Processing of the results is also completely flexible. However, the modelling concepts in DAE Tools cannot be expressed directly in the programming language and must be emulated in its API. Also, it is programming language dependent. To certain extent, this can be overcome by the fact that Python shines as a glue language, used to combine components written in different programming languages and a large number of scientific software libraries expose its functionality to Python via their extension modules.

Regarding the available modelling techniques, three approaches currently exist (*Morton, 2003*): (a) sequential modular, (b) simultaneous modular, and (c) equation-based (acausal). The equation-based approach is adopted and implemented in this work. A brief history of the equation-based solvers and comparison of the sequential-modular and equation-based approaches can be found in *Morton (2003)* and a good overview of the equation-oriented approach and its application in gPROMS is given by *Barton & Pantelides (1993)*. According to this approach, all equations and variables which constitute the model representing the process are generated and gathered together. Then, equations are solved simultaneously using a suitable mathematical algorithm (*Morton, 2003*). In the equation-based approach, equations are given in an implicit form as functions of state variables and their derivatives, degrees of freedom (the system variables that may vary independently), and parameters:

$$F(\dot{x}, x, y, p) = 0$$

where $x$ represents state variables, $\dot{x}$ their derivatives, $y$ degrees of freedom and $p$ parameters. Input-output causality is not fixed providing a support for different simulation scenarios (based on a single model) by fixing different degrees of freedom.

The hybrid approach allows an easy interaction with other software packages/libraries. First, other numerical libraries can be accessed directly from the code, and since the Python's design allows an easy development of extension modules from different languages, a vast number of numerical libraries is readily available. Second, DAE Tools are developed with a built-in support for NumPy (http://numpy.scipy.org) and SciPy (http://scipy.org) numerical packages; therefore, DAE Tools objects can be used as native NumPy data types and numerical functions from other extension modules can directly operate on them. This way, a large pool of advanced and massively tested numerical algorithms is made directly available to DAE Tools.

The automatic differentiation is always utilised to analytically generate the Jacobian matrix if the direct sparse linear solvers are used, or to generate a preconditioner matrix for the iterative linear solvers using the software suites such as Trilinos AztecOO, IFPACK, and ML. The automatic differentiation is also applied to sensitivity analysis where it is used to calculate derivatives of model equations per parameters with respect to which sensitivities are requested. Only the continuous-time systems are supported and the forward sensitivity method provided by the Sundials IDAS solver is available at the moment. The forward sensitivity equations are integrated together with the original DAE system leading to the

DAE system of size $N(N_s + 1)$, where $N$ is the size of the original DAE system and $N_s$ is the number of model parameters. More information about the sensitivity analysis using the forward sensitivity method can be found in the Sundials documentation.

DAE Tools also provide code generators and co-simulation/model exchange standards/interfaces for other simulators. This way, the developed models can be simulated in other simulators either by generating the source code, exporting a model specification file or through some of the standard co-simulation interfaces. To date, the source code generators for c99, Modelica and gPROMS languages have been developed. In addition, DAE Tools functionality can be exposed to MATLAB, Scilab and GNU Octave via MEX-functions, to Simulink via user-defined S-functions and to the simulators that support FMI co-simulation capabilities. The future work will concentrate on support for the additional interfaces (i.e. CAPE-OPEN) and development of additional code generators.

Parallel computation is supported using only the shared-memory parallel programming model at the moment. Since a repeated solution of the system of linear equations typically requires around 90–95% of the total simulation time, the linear equations solver represents the major bottleneck in the simulation. Therefore, the main focus was put on performance improvement of the solution of linear equations using one of the available multi-threaded solvers such as SuperLU_MT, Pardiso and Intel Pardiso.

## ARCHITECTURE

DAE Tools consists of six packages: *core*, *activity*, *solvers*, *datareporting*, *logging*, and *units*. All packages provide a set of interfaces (abstract classes) that define the required functionality. Interfaces are realised by the implementation classes. The implementation classes share the same name with the interface they realise with the suffix *_t* dropped (i.e. the class *daeVariable* implements interface *daeVariable_t*).

### Package "core"

This package contains the key modelling concepts. The class diagram with interfaces (abstract classes) is presented in Fig. S1. The most important modelling concepts are given in Table 1. Interface realisations are given in Fig. S2. Models in DAE Tools are represented by the *daeModel* class and contain the following elements: domains, parameters, variables, equations, state transition networks, ports, event ports, actions to be performed when a given condition is satisfied, actions to be performed when an event is triggered on a given event port, and components (instances of other models, used to form a hierarchy of models). The *daeModel* UML class diagram is presented in Fig. S3.

### Package "activity"

This package contains interfaces that define an API for activities that can be performed on developed models. To date, only two interfaces are defined and implemented: *daeSimulation_t* (defines a functionality used to perfom simulations) and *daeOptimization_t* (defines a functionality used to perform optimisations).

**Table 1** The key modelling concepts in DAE Tools software.

| Concept | Description |
| --- | --- |
| *daeVariableType_t* | Defines a variable type that has the units, lower and upper bounds, a default value and an absolute tolerance |
| *daeDomain_t* | Defines ordinary arrays or spatial distributions such as structured and unstructured grids; parameters, variables, equations and even models and ports can be distributed on domains |
| *daeParameter_t* | Defines time invariant quantities that do not change during a simulation, such as a physical constant, number of discretisation points in a domain etc. |
| *daeVariable_t* | Defines time varying quantities that change during a simulation |
| *daePort_t* | Defines connection points between model instances for exchange of continuous quantities; similar to the models, ports can contain domains, parameters and variables |
| *daeEventPort_t* | Defines connection points between model instances for exchange of discrete messages/events; events can be triggered manually or when a specified condition is satisfied; the main difference between event and ordinary ports is that the former allow a discrete communication between models while latter allow a continuous exchange of information |
| *daePortConnection_t* | Defines connections between two ports |
| *daeEventPortConnection_t* | Defines connections between two event ports |
| *daeEquation_t* | Defines model equations given in an implicit/acausal form |
| *daeSTN_t* | Defines state transition networks used to model discontinuous equations, that is equations that take different forms subject to certain conditions; symmetrical/non-symmetrical and reversible/irreversible state transitions are supported |
| *daeOnConditionActions_t* | Defines actions to be performed when a specified condition is satisfied |
| *daeOnEventActions_t* | Defines actions to be performed when an event is triggered on the specified event port |
| *daeState_t* | Defines a state in a state transition network; contains equations and on_event/condition action handlers |
| *daeModel_t* | Represents a model |

## Package "solvers"

This package contains interfaces that define an API for numerical solution of systems of Differential Algebraic Equations (DAE), systems of Linear Equations (LA), and (mixed-integer) nonlinear programming problems (NLP or MINLP), and auxiliary classes. The class diagram with the defined interfaces is presented in Fig. S4: *daeDAESolver_t* (defines a functionality for the solution of DAE systems), *daeNLPSolver_t* (defines a functionality for the solution of (MI)NLP problems), *daeLASolver_t* (defines functionality for the solution of systems of linear equations) and *daeIDALASolver_t* (derived from *daeLASolver_t,* used by Sundials IDAS linear solvers). Interface realizations are given in Fig. S5. Current implementations include Sundials IDAS DAE solver, IPOPT, BONMIN and NLOPT (MI) NLP solvers and SuperLU, SuperLU_MT, PARDISO, Intel PARDISO and Trilinos (Amesos and AztecOO) sparse matrix linear solvers. Since all these linear equation solvers use different sparse matrix representations, a generic interface (template *daeMatrix<typename FLOAT>*) has been developed for the basic operations performed by DAE Tools software such as setting/getting the values and obtaining the matrix properties. This way, DAE Tools objects can access the matrix data in a generic fashion while hiding the internal implementation details. To date, three matrix types have been implemented: *daeDenseMatrix, daeLapackMatrix* (basically wrappers around C/C++ and Fortran two-dimensional arrays), a template class *daeSparseMatrix<typename FLOAT, typename INT>* (sparse matrix) and its realization *daeCSRMatrix<typename FLOAT, typename INT>* implementing the Compressed Row Storage (CSR) sparse matrix representation.

## Package "datareporting"

This package contains interfaces that define an API for processing of simulation results by the *daeSimulation_t* and *daeDAESolver_t* classes, and the data structures available to access those data by the users. Two interfaces are defined: *daeDataReporter_t* (defines a functionality used by a simulation object to report the simulation results) and *daeDataReceiver_t* (defines a functionality/data structures for accessing the simulation results). A number of data reporters have been developed for: (a) sending the results via TCP/IP protocol to the DAE Tools Plotter application (*daeTCPIPDataReporter*), (b) plotting the results using the Matplotlib Python library (*daePlotDataReporter*), and (c) exporting the results to various file formats (such as MATLAB MAT, Microsoft Excel, html, xml, json and HDF5). An overview of the implemented classes is given in Fig. S6.

## Package "logging"

This package contains only one interface *daeLog_t* that define an API for sending messages from the simulation to the user. Interface realizations are given in Fig. S7. Three implementations exist: *daeStdOutLog* (prints messages to the standard output), *daFileLog* (stores messages to the specified text file), and *daeTCPIPLog* (sends messages via TCP/IP protocol to the *daeTCPIPLogServer*; used when a simulation is running on a remote computer).

## Package "units"

Parameters and variables in DAE Tools have a numerical value in terms of a unit of measurement (quantity) and units-consistency of equations and logical conditions is strictly enforced (although it can be switched off, if required). The package contains only two classes: *unit* and *quantity*. Both classes have overloaded operators +, −, *, / and ** to support creation of derived units and operations on quantities that contain a numerical value and units. In addition, the package defines the basic mathematical functions that operate on *quantity* objects (such as *sin, cos, tan, sqrt, pow, log, log10, exp, min, max, floor, ceil, abs* etc.).

## SOLUTION OF A DAE SYSTEM

The solution of a DAE system requires the functionality provided by the following objects: (a) simulation object implementing the *daeSimulation_t* interface (*simulation*), (b) DAE solver object implementing the *daeDAESolver_t* interface (*dae_solver*), (c) linear equations solver object implementing the *daeLASolver_t* interface (*la_solver*), (d) data reporter object implementing the *daeDataReporter_t* interface (*data_reporter*), and (e) log object implementing the *daeLog_t* interface (*log*). A diagram illustrating the participating objects and their associations are given in Fig. 1. Solution of an optimisation problem includes an identical set of objects with the addition of *optimization* object implementing *daeOptimization_t* interface and *nlp_solver* object implementing the *daeNLPSolver_t* interface. Solution of a DAE system is performed in five phases: (I) creation and initialisation of objects in the main program, (II) initialisation of the simulation and runtime checks in *daeSimulation::Initialize()* function, (III) calculation of initial conditions

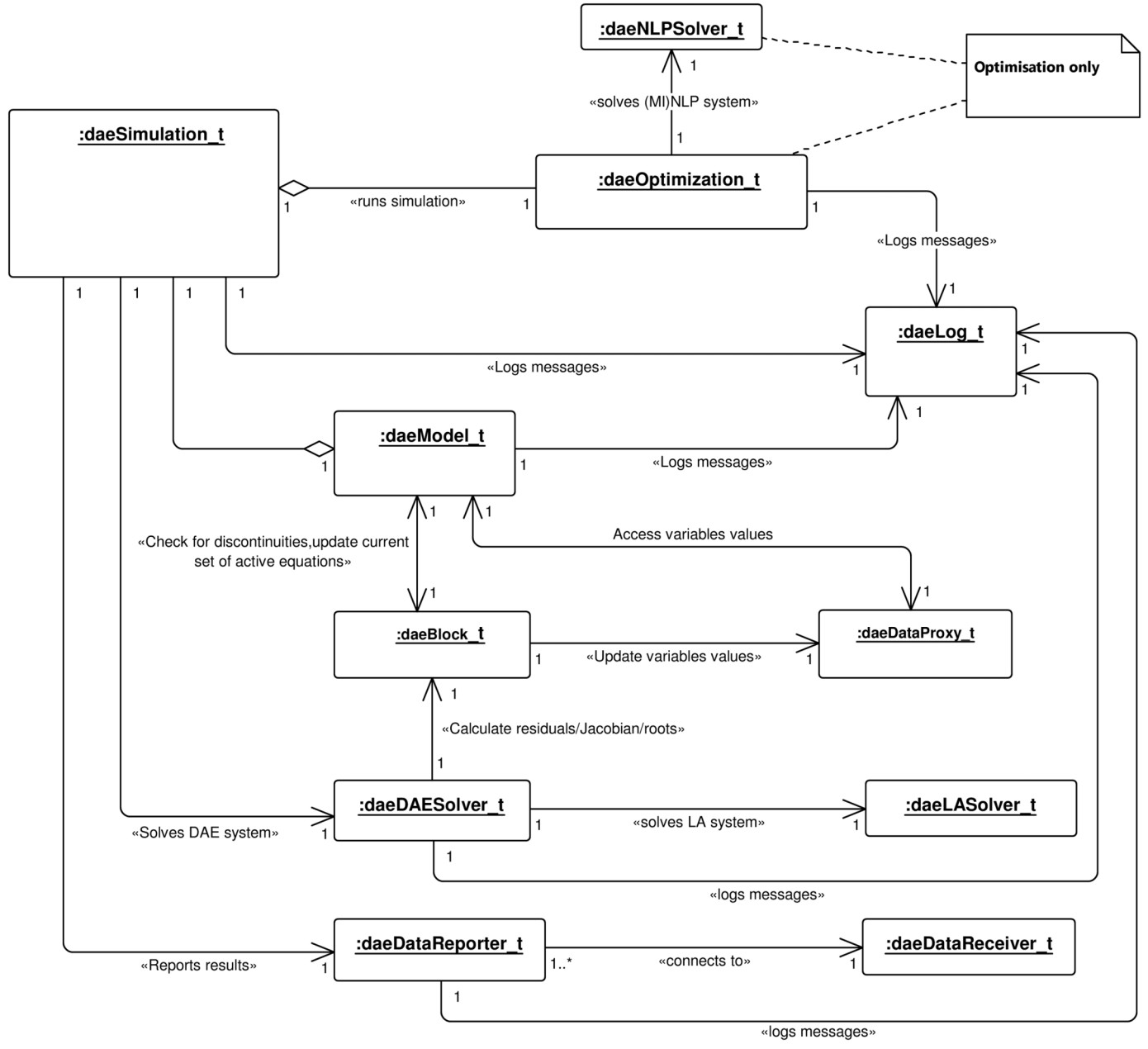

**Figure 1** **UML object diagram.** DAE Tools simulation/optimisation.

in *daeSimulation::SolveInitial()* function, (IV) integration of the DAE system in time in *daeSimulation::Run()* function, and (V) clean-up in *daeSimulation::Finalize()* function followed by destruction of objects in the main program. A typical sequence of calls during the DAE Tools simulation are given in Fig. 2.

### Phase I: creation of objects

*simulation*, *dae_solver*, *la_solver*, *data_reporter* and *log* objects are instantiated in the main program. All distribution domains, parameters, variables and ports are now instantiated.

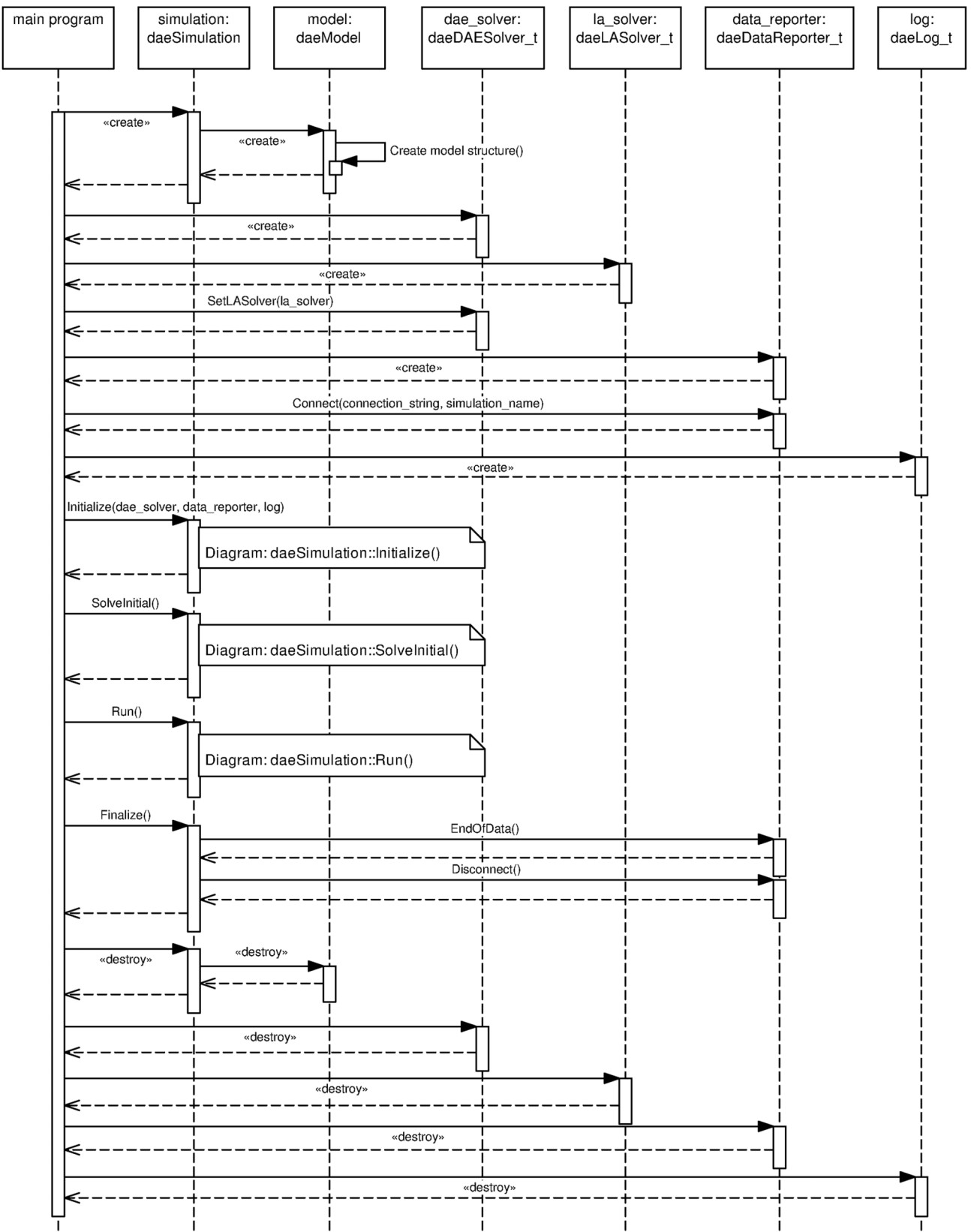

**Figure 2 UML sequence diagram.** DAE Tools simulation.

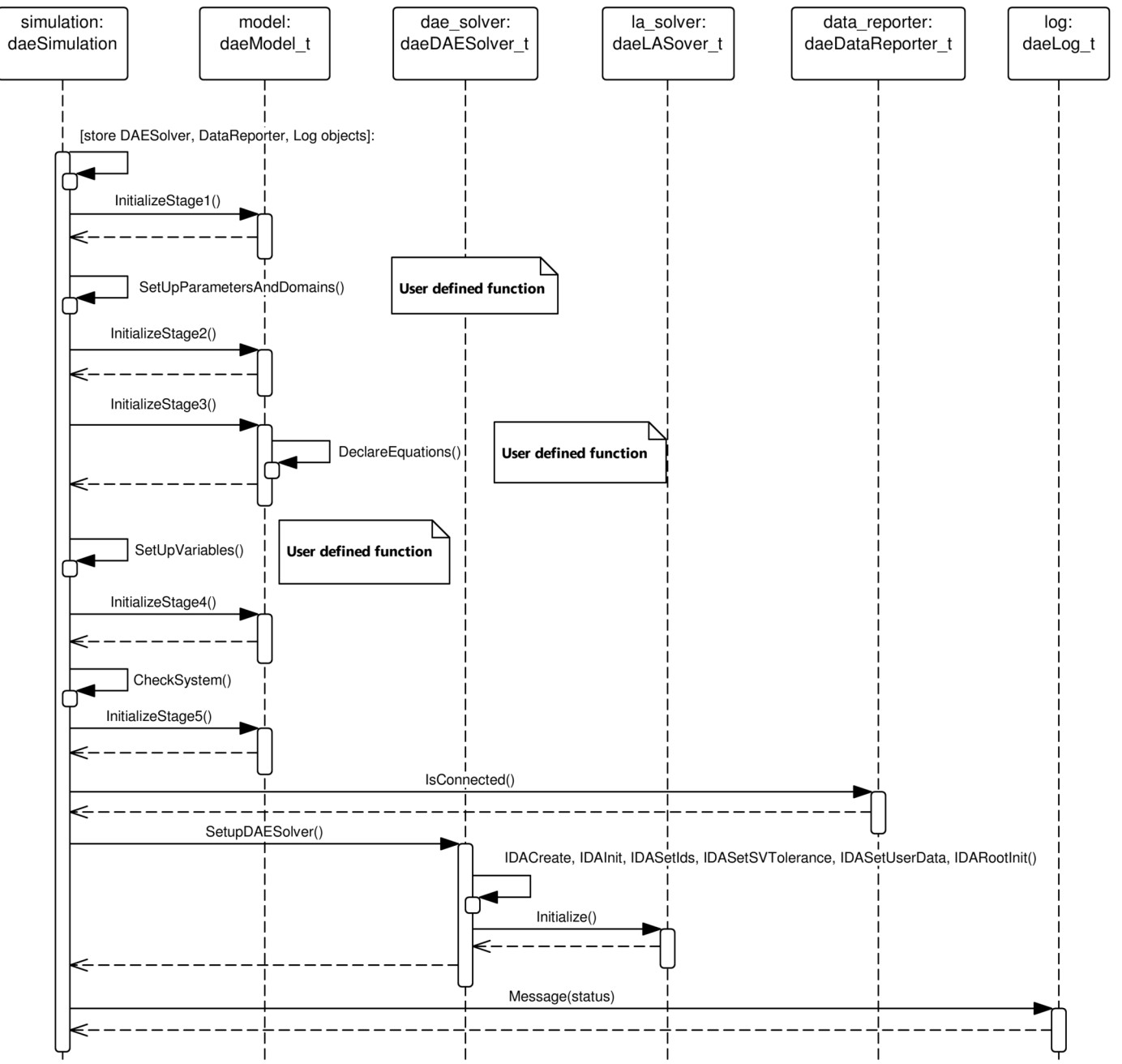

**Figure 3  UML sequence diagram.** *daeSimulation::Initialize()* function.

## Phase II: initialisation and runtime checks

A sequence of calls during the initialisation in *daeSimulation:Initialize()* function is given in Fig. 3. During stage 1, *daeDataProxy_t* instance is created. In DAE Tools approach, variables do not hold the values the values are stored in a proxy object as a compact block of memory to improve the memory copy performance. A separate storage is created for state variables and their derivatives and for degrees of freedom.

The variables access the data using their global index. The user-defined function *SetUpParametersAndDomains()* from the *daeSimulation*-derived class is now called where the parameters values are assigned and the distribution domains initialised. In stage 2 of initialisation, the port and model arrays are created and every variable obtains assigned the global index. Distributed variables obtain a separate index for every point in domains they are distributed on. In stage 3, based on the number of variables and their types, the memory storage for variables values and derivatives is allocated in the data proxy object, the user-defined function *DeclareEquations()* from the *daeModel*-derived classes called to create equations, state transition networks, port connections and *OnCondition*/*OnEvent* handlers, and the initial variables values and absolute tolerances are set. In the stage 4, the equations get initialised and expanded into an array of residual expressions (one for every point in domains that the equation is distributed on). Every residual expression is evaluated to form an evaluation tree. The concept of representing equations as evaluation trees is employed for evaluation of residual equations and their gradients (which represent a single row in the Jacobian matrix). This is achieved by using the operator overloading technique for automatic differentiation adopted from the ADOL-C library (*Walther & Griewank, 2012*). Evaluation trees consist of unary and binary nodes, each node representing a parameter/variable value, basic mathematical operation (+, −, *, /, **) or a mathematical function (*sin, cos, tan, arcsin, arccos, arctan, sinh, cosh, tanh, arcsinh, arccosh, arctanh, arctan2, erf, sqrt, pow, log, log10, exp, min, max, floor, ceil, abs, sum, product, integral*, etc.). The mathematical functions are overloaded to operate on a heavily modified ADOL-C class *adouble*, which has been extended to contain information about domains, parameters and variables. In adition, a new *adouble_array* class has been introduced to support the above-mentioned operations on arrays of parameters and variables. Once built, the evaluation trees can be used for several purposes: (a) to calculate equation residuals, (b) to calculate equation gradients, (c) to export equation expressions into the MathML or LaTeX format, (d) to generate the source code for different languages, and (e) to perform various types of runtime checks. A typical evaluation tree is presented in Fig. 4. In stage 5, the *daeBlock* instance is created which is used by a DAE solver during the integration of the DAE system. It represents a block of equations and holds the currently active set of equations (including those from state transition networks) and root functions. Finally, the whole system is checked for errors/inconsistencies and the DAE solver initialised.

### Phase III: calculation of initial conditions

A sequence of calls during the calculation of initial conditions in *daeSimulation: SolveInitial()* function is given in Fig. 5. The consistent set of initial conditions is obtained using the *IDACalcIC()* function which repeatedly calls the functions to evaluate equations residuals, Jacobian matrix and root functions, solves the resulting system of linear equations and checks for possible occurrences of discontinuities until the specified tolerance is achieved.

$$F = \frac{dx_1}{dt} + \frac{x_2}{x_3 + 2.5} + sin(x_4) = 0$$

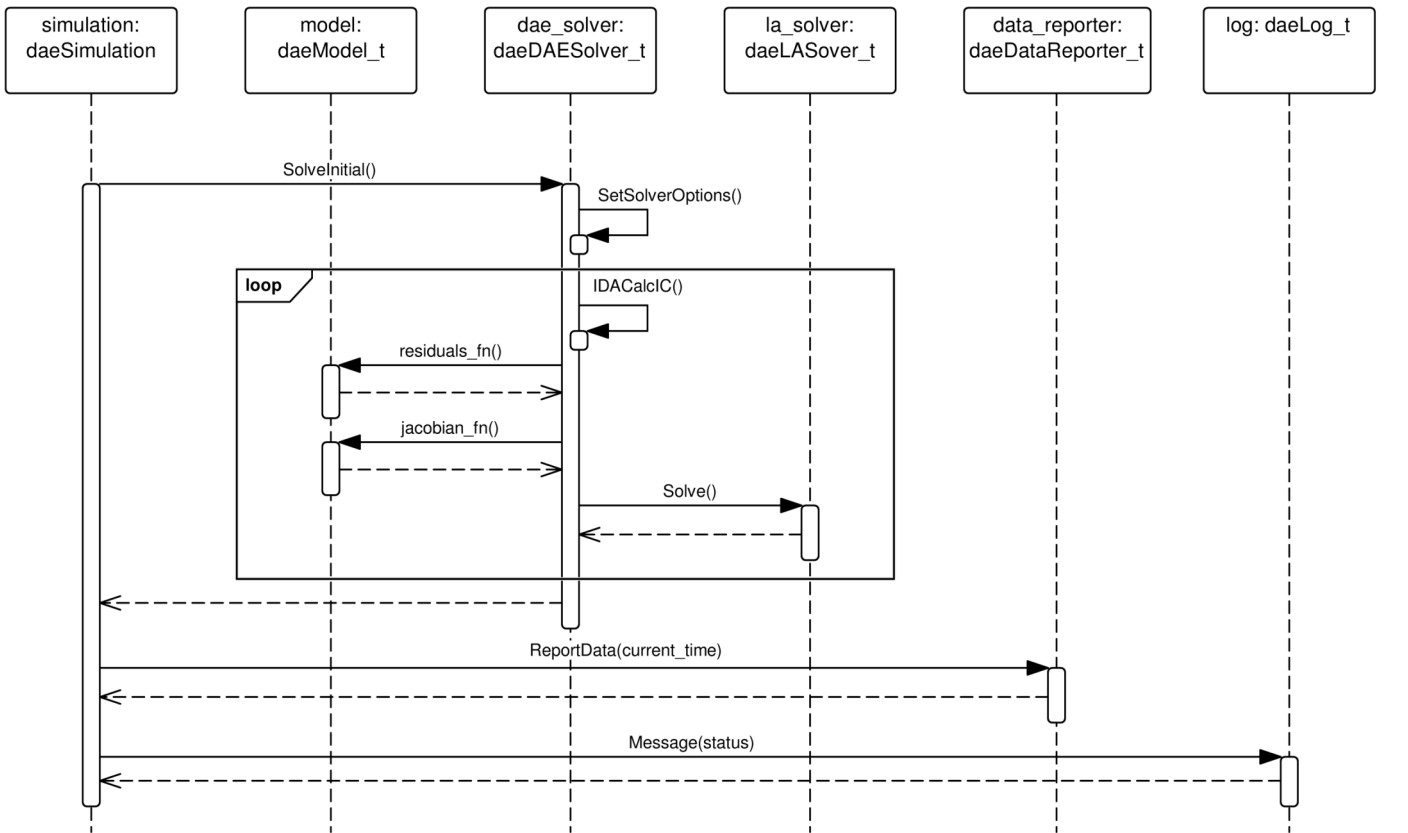

**Figure 4 Equation evaluation tree.**

**Figure 5 UML sequence diagram.** *daeSimulation::SolveInitial()* function.

### Phase IV: integration in time

A sequence of calls during the integration of the system in *daeSimulation::Run()* function is given in Fig. 6. The default implementation calls *daeSimulation::IntegrateUntilTime()* and *daeSimulation::ReportData()* functions in a loop until the specified time horizon is

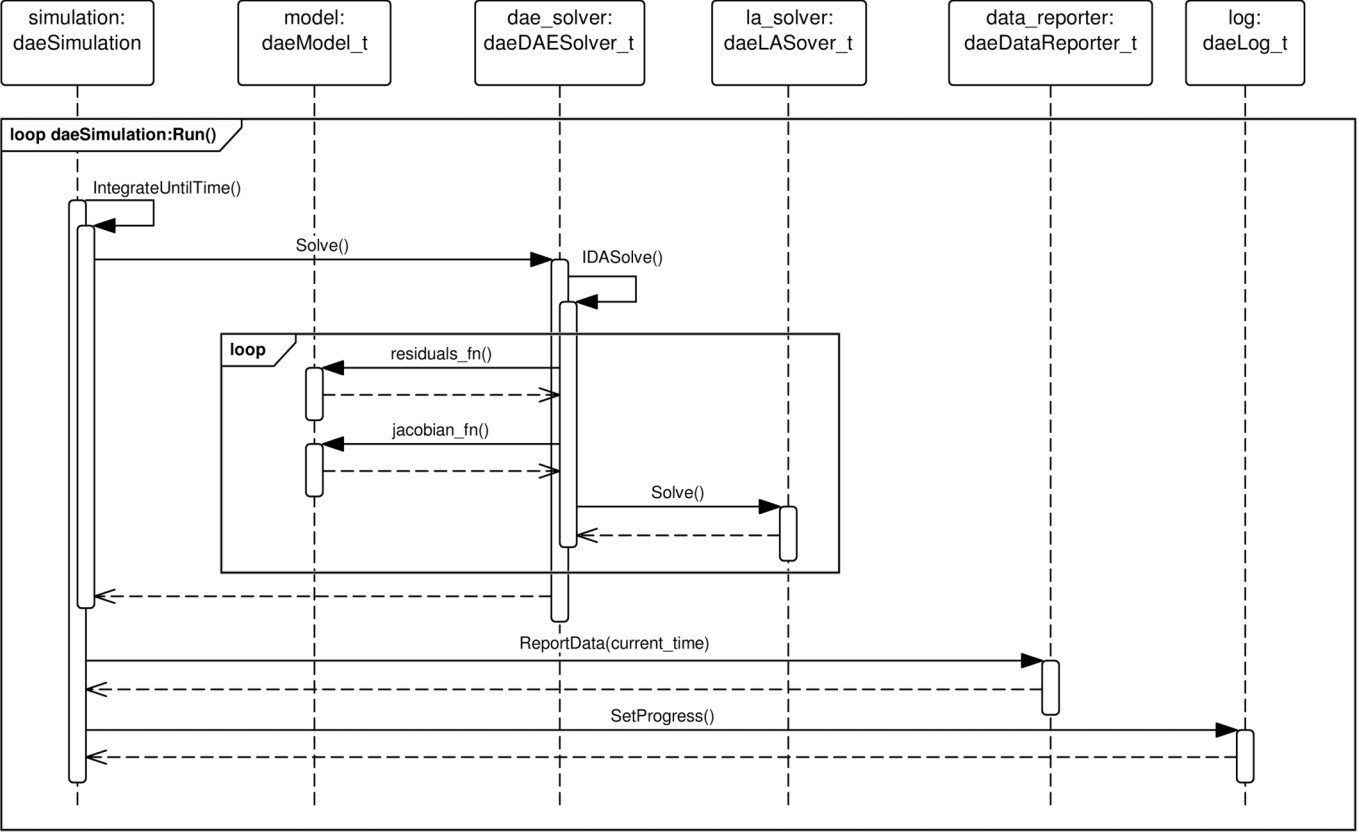

**Figure 6 UML sequence diagram.** *daeSimulation::Run()* function.

reached. The *IntegrateUntilTime()* function uses the *IDASolve()* function that repeatedly calls the functions to evaluate equations residuals, Jacobian matrix and root functions, solves the resulting system of linear equations and checks for possible occurrences of discontinuities until the specified tolerance is achieved.

### Phase V: clean up

This phase includes a call to *daeSimulation::Finalize()* function which performs internal clean-up and memory release, followed by destruction of objects instantiated during phase I.

## DEVELOPING MODELS WITH DAE TOOLS

In DAE Tools, models are developed by deriving new classes from the base model class (*daeModel*). The procedure consists of two steps: (a) declaration of the model structure (domains, parameters, variables, ports etc.) in the *__init__()* function (from Python) or in the *daeModel*-derived class constructor (from C++), (b) specification of the model functionality (equations and state transition networks) in the *DeclareEquations()* function. A simple model developed using the Python programming language is given in Source Code Listing 4 and the same model developed in C++ in the Supplemental Source Code Listing S2. More information about the API, the user guide and tutorials can be found in the

*Documentation* section of the DAE Tools website (http://www.daetools.com/docs/index.html), subsections *pyDAE User Guide*, *pyDAE API Reference*, and *Tutorials*, respectively. The model describes a block of copper at one side exposed to the source of heat and at the other to the surroundings with the constant temperature and the constant heat transfer coefficient. The process starts at the temperature of the metal of 283 K. The integral heat balance can be described by the following ordinary differential equation:

$$mc_p \frac{dT}{dt} = Q_{in} - \alpha A(T - T_{surr})$$

where $m$ is a mass of the block, $c_p$ is the specific heat capacity, $T$ is the temperature, $Q_{in}$ is the input power of the heater, $\alpha$ is the heat transfer coefficient, $A$ is the surface area of the block and $T_{surr}$ is the temperature of the surroundings. The copper block model is simulated for 500 s. At a certain point in time, the heat produced by the heater becomes equal to the heat removed by natural convection and the system reaches the steady-state.

**Listing 4 CopperBlock model (Python)**

```python
from daetools.pyDAE import *
from pyUnits import m, kg, s, K, Pa, J, W

# Part 1: creating a model
class CopperBlock(daeModel):
    def __init__(self, Name, Parent = None, Description = ""):
        daeModel.__init__(self, Name, Parent, Description)

        self.m     = daeParameter("m",       kg,          self, "Mass of the copper block")
        self.cp    = daeParameter("c_p",     J/(kg*K), self, "Specific heat capacity")
        self.alpha = daeParameter("α", W/((m**2)*K), self, "Heat transfer coefficient")
        self.A     = daeParameter("A",       m**2, self, "Surface area for the heat transfer")
        self.Tsurr = daeParameter("T_surr",  K,    self, "Temperature of the surroundings")

        self.Qin  = daeVariable("Q_in", power_t,       self, "Power of the heater")
        self.T    = daeVariable("T",    temperature_t, self, "Block temperature")

    def DeclareEquations(self):
        daeModel.DeclareEquations(self)

        eq = self.CreateEquation("HeatBalance", "Integral heat balance equation")
        eq.Residual = self.m() * self.cp() * self.T.dt() - self.Qin() + \
                      self.alpha() * self.A() * (self.T() - self.Tsurr())
```

Definition of a simulation in DAE Tools requires the following steps: (a) deriving a new class from the base simulation *daeSimulation* class; (b) specification of the model to be simulated in the *__init__()* function (from Python) or in *daeSimulation*-derived class constructor (from C++); (c) setting the values of parameters and distribution domains in *SetUpParametersAndDomains()* function; (d) fixing the degrees of freedom by assigning the values to certain variables, setting the initial conditions for differential variables and other information such as initial guesses, absolute tolerances, etc. in *SetUpVariables()*; (e) specification of an operating procedure in *Run()* function (it can be either a simple run

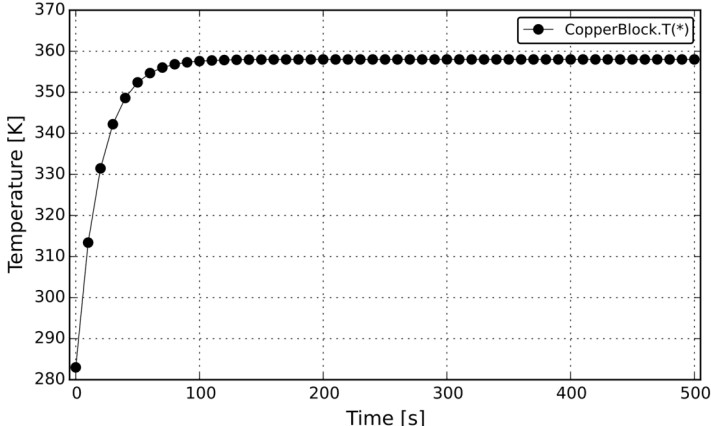

**Figure 7 Temperature profile from the CopperBlock simulation.**

for a specified period of time which is default, or a complex one where various actions can be taken during the simulation). The simulation classes for the copper block model are shown in Source Code Listing 5 (Python).

---

**Listing 5 CopperBlock simulation (Python)**

```python
# Part 2: setting up a simulation
class simCopperBlock(daeSimulation):
    def __init__(self):
        daeSimulation.__init__(self)
        self.m = CopperBlock("CopperBlock")

    def SetUpParametersAndDomains(self):
        self.m.cp.SetValue(385 * J/(kg*K))
        self.m.m.SetValue(1 * kg)
        self.m.alpha.SetValue(200 * W/((m**2)*K))
        self.m.A.SetValue(0.1 * m**2)
        self.m.Tsurr.SetValue(283 * K)

    def SetUpVariables(self):
        # Set input power of the heater
        # It is a degree of freedom (DOF) and must be assigned
        self.m.Qin.AssignValue(1500 * W)

        # Set an initial condtion for the temperature
        self.m.T.SetInitialCondition(283 * K)
```

---

Running a simulation requires the following steps: (a) instantiation of DAE and LA solvers, data reporter, data receiver, and log objects; (b) setting a time horizon and a reporting interval; (c) initialisation of the simulation; (d) calculation of the initial conditions; (e) running simulation; and (f) cleaning up. The simulation performed in Python is given in the Source Code Listing 6. The complete source code of the copper block models developed in Python and C++ are given in the Supplemental Listings S1 and S2, respectively. The simulation results for the copper block model are presented in Fig. 7.

**Listing 6 Running the CopperBlock simulation (Python)**

```python
# Part 3: running a simulation
# Create Log, Solver, DataReporter and Simulation object
log           = daePythonStdOutLog()
daesolver     = daeIDAS()
datareporter  = daeTCPIPDataReporter()
simulation    = simCopperBlock()

# Enable reporting of all variables
simulation.m.SetReportingOn(True)

# Set the time horizon and the reporting interval
simulation.ReportingInterval = 10
simulation.TimeHorizon = 500

# Connect the TCP/IP data reporter (the default address is "localhost:50000")
datareporter.Connect("", "CopperBlock")

# Initialize the simulation
simulation.Initialize(daesolver, datareporter, log)
# Solve the system at time = 0
simulation.SolveInitial()
# Run the simulation
simulation.Run()
# Clean-up
simulation.Finalize()
```

The previous example was very simple. DAE Tools also support some advanced features such as discontinuous equations and state transition networks. More information about state transition networks and their types can be found in *Barton & Pantelides (1993)*. Consider a slightly more complex problem now: the same copper block at the ambient temperature (283 K) is allowed to warm up for 200 s, the heat source is then switched off and the metal cools down to the ambient temperature. This problem can be modelled using the concept of symmetrical State Transition Networks (STN); in DAE Tools this type of STN can be created using *IF*, *ELSE_IF*, *ELSE*, and *END_IF* functions from the *daeModel* class. From the modellers perspective, these function behave in a similar fashion as ordinary *if-else_if-else* blocks in all programming languages and select the active set of equations based on the specified logical conditions. The source code of this model is given in Source Code Listing 7 and the simulation results in Fig. 8. The complete source code of the modifed model is given in Supplemental Listing S3.

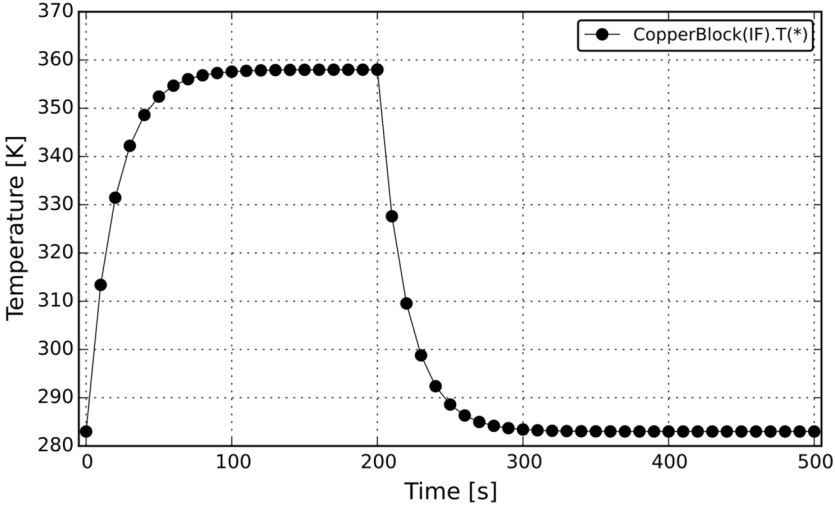

**Figure 8 Temperature profile from the CopperBlock simulation (symmetrical reversible STN).**

---

**Listing 7 CopperBlock model with the symmetrical reversible STN (Python)**

```python
class CopperBlock(daeModel):
    ...
    def DeclareEquations(self):
        ...
        self.IF(Time() < Constant(200*s))
        eq = self.CreateEquation("Q_on", "The heater is on")
        eq.Residual = self.Qin() - Constant(1500 * W)

        self.ELSE()
        eq = self.CreateEquation("Q_off", "The heater is off")
        eq.Residual = self.Qin()

        self.END_IF()
```

Another commonly used type of state transition networks is a non-symmetrical STN. This type of STN in DAE Tools can be created by using *STN*, *STATE*, and *END_STN* functions from the *daeModel* class. To illustrate this concept, consider again the same copper block problem. The process starts again at the temperature of 283 K. However, this time the temperature of the copper block is allowed to reach 340 K and once that is done its temperature is kept in the interval between 320–340 K for 350 s by switching the heater on and off. After 350 s, the heat source is permanently switched off and the block cools down to the ambient temperature. The source code of the new model is given in Source Code Listing 8 and the simulation results in Fig. 9. The complete source code is given in Supplemental Listing S4.

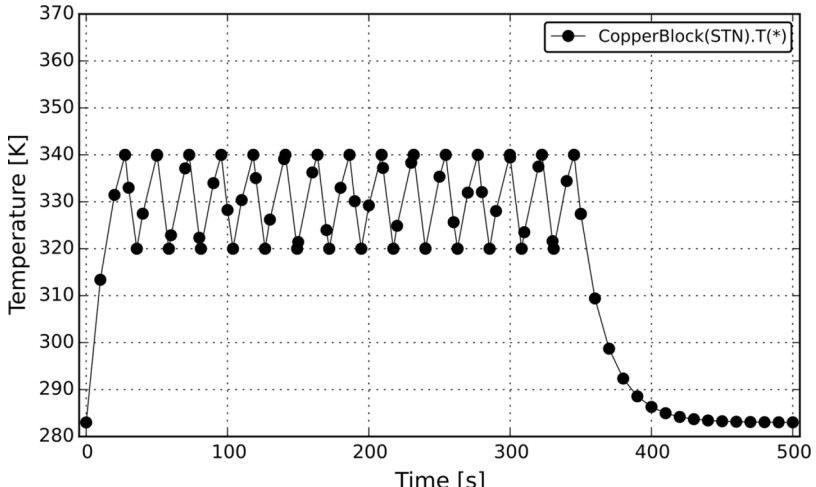

**Figure 9** Temperature profile from the CopperBlock simulation (non-symmetrical irreversible STN).

---

**Listing 8  CopperBlock model with the non-symmetrical irreversible STN (Python)**

```python
class CopperBlock(daeModel):
    ...
    def DeclareEquations(self):

        ...
        self.stnRegulator = self.STN("Regulator")
        self.STATE("HeaterOn")
        eq = self.CreateEquation("HeaterInput")
        eq.Residual = self.Qin() -  Constant(1500 * W)
        self.ON_CONDITION(self.T() > Constant(340*K),
                              switchToStates = [ ('Regulator', 'HeaterOff') ] )
        self.ON_CONDITION(Time() > Constant(350*s),
                              switchToStates = [ ('Regulator', 'RegulatorOff') ] )

        self.STATE("HeaterOff")
        eq =  self.CreateEquation("HeaterInput")
        eq.Residual = self.Qin()
        self.ON_CONDITION(self.T() < Constant(320*K),
                              switchToStates = [ ('Regulator','HeaterOn') ])
        self.ON_CONDITION(Time() > Constant(350*s),
                              switchToStates = [ ('Regulator', 'RegulatorOff') ] )

        self.STATE("RegulatorOff")
        eq =  self.CreateEquation("HeaterInput")
        eq.Residual = self.Qin()

        self.END_STN()
```

## APPLICATIONS

To date, DAE Tools have been applied to several diverse scientific areas such as gas adsorption, porous membranes, crystallisation, electrochemistry and biological neural networks. Two projects utilising DAE Tools software at different levels of abstraction are presented in this work. The first example illustrates applicability of DAE Tools to development of multi-scale models. In this example, the authors took advantage of the built-in interoperability with Python NumPy library to perform vector operations on NumPy arrays to form the DAE system. The second example illustrates two important DAE Tools capabilities: (a) embedding into another software (a domain specific language simulator) running on a server and providing its functionality through a web service or web application, and (b) defining the modelling concepts from a new application domain using the DAE Tools fundamental modelling concepts.

### Multi-scale model of phase-separating battery electrodes

In the work of *Li et al. (2014)*, DAE Tools has been applied to modelling of lithium-ion batteries. Lithium-ion batteries operate by shuttling lithium ions from one electrode to the other. In a charged state, the lithium ions are stored in the negative electrode (anode), and the positive electrode (cathode) has almost no lithium. During a discharge, lithium exits the anode, sending an electron through the outer circuit. Lithium ions then move through an electrolyte phase to the cathode, where they recombine with an electron as they enter the cathode. To correctly describe the physics of this process, transport of lithium within the electrodes and within the electrolyte has to be modelled as well as the electrochemical reactions in which lithium ions separate/combine with an electron to exit/enter the electrode materials. Complicating the modelling process, battery electrodes are typically made out of a porous material composed of large numbers of small, solid active particles with a percolating electrolyte. This provides a large surface area for electrochemical reactions to drive electrons through the outer circuit but also creates a strong separation of length scales. The electrode may have a typical thickness of hundreds of microns, whereas single electrode particles range from tens of nanometers to tens of microns. In addition, the system inherently has highly separated time scales. Particles may have transport time scales less than one second, but the imposed time scale for battery discharge is typically on the order of hours. One approach to simulating this system is referred to as porous electrode theory. Porous electrode theory for battery simulations is a method of systematically coupling the different length scales and physical phenomena involved in battery operation. The basic approach involves writing conservation equations both for lithium transport within the particles (small length scale) and for lithium ion transport through the electrolyte (large length scale). Directly simulating the full micro-structure of the electrode particles and electrolyte pores within the porous electrode would require enormous computational effort. Instead, the two phases are coupled via a volume-averaged approach in which simulated particles act as volumetric source/sink terms as they interact with the electrolyte via reactions. More details about the governing equations of such a model applied to a battery electrode made of $LiFePO_4$ can be found in *Li et al. (2014)* and its

Supplemental Information. Spatial discretisation of the governing equations is carried out using the finite volume method, as solid particles are described as residing within individual electrode volumes, as depicted in Fig. 10.

The resulting discretised set of equations is a large system of DAE's. Differential equations come from the discretised transport equations, and algebraic constraints arise from electrostatic equations and constraints on the total integrated reaction rate (current). In *Li et al. (2014)*, the discretised system of DAE's was integrated in time using MATLAB's ode15s solver and subsequently reimplemented using DAE Tools, allowing a direct comparison between the two integrators. Using default solver tolerances for both ($10^{-3}$ in MATLAB and $10^{-5}$ in DAE Tools), a number of simulations were carried out using both the MATLAB implementation and the DAE Tools implementation, and in each case the simulation outputs were indistinguishable. Despite obtaining equivalent outputs, the implementation using DAE Tools consistently ran more quickly (Fig. 11), up to ten times faster (4.22 times, in average). This speedup is a result of its built-in support for automatic differentiation facilitating rapid and accurate derivative evaluation for solution of the highly non-linear system of equations involved in time stepping. In contrast, the ode15s solver creates a numerical approximation of the Jacobian matrix if the Jacobian calculation function is not provided as an input; therefore, the convergence rate is much slower. The significant loss in performance illustrates the benefits of the object-oriented DAE Tools API and the automatic differentiation capabilities it provides, since the calculation of derivatives by hand for all functions in the MATLAB model is very difficult and error-prone for a system of this size and complexity. In addition, the equation-oriented modelling approach made the implementation both easier to code and easier to read in comparison to the non-intuitive mass-matrix approach for coupled differential variables used in the MATLAB version. Finally, the object-oriented approach and clear model separation of DAE Tools also facilitates a much more maintainable and extensible code base in which particle models can be easily interchanged, added, and incorporated into other electrode models.

## NineML domain specific language

DAE Tools software has been used as a reference implementation simulator for the "Network Interchange format for NEuroscience" (NineML) Modelling Language. NineML is an open source xml-based domain specific language for modelling of networks of spiking neurones. It is a simulator-independent language with the aim of providing an unambiguous description of neuronal network models for efficient model sharing and reusability between different simulators (such as NEURON, NEST, NeuroML/LEMS etc.). The language has emerged from a joint effort of experts in the fields of computational neuroscience, simulator development and simulator-independent language initiatives (NeuroML, PyNN), grouped in the INCF Multiscale Modelling Task Force (http://www.incf.org). NineML consists of two layers: an Abstraction Layer (AL) contains mathematical description and concepts, and a User Layer (UL) contains parameters values and instantiations. The key modelling concepts in the language are: (a) cell models (spiking neurons); (b) synapse models; (c) groups of neurons such as populations

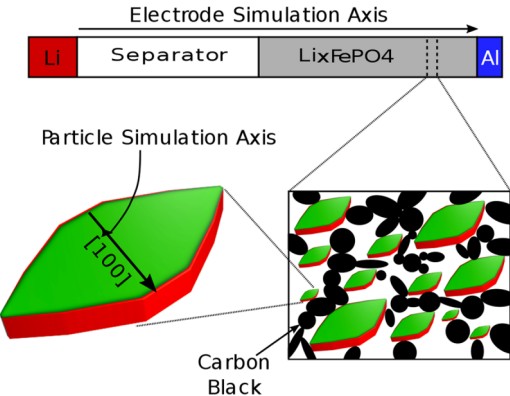

**Figure 10  Schematic of the multi-scale porous electrode model.**

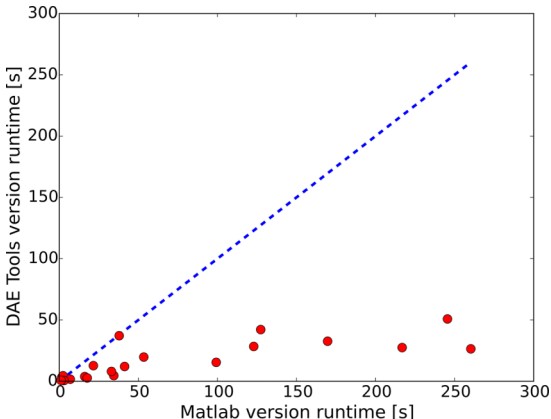

**Figure 11  Parity plot for simulation runs with different inputs (MATLAB vs. DAE Tools).**

and layers; and (d) connectivity patterns–projections, connection probabilities, etc. Apart from the specification, the language contains a set of additional tools: (a) Python library (lib9ml); (b) abstraction layer component tester and report generator (DAE Tools based); and (c) the reference implementation simulator (DAE Tools based). The source code and more information about the language and the whole project can be found on the INCF's portal (http://software.incf.org/software/nineml).

### *Abstraction layer component tester and report generator*

The purpose of the application is validation and testing of abstraction layer components and generation of model reports (as a help to components developers). The application is available in three flavours: (a) desktop application with the pyQt graphical user interface; (b) web application with the jQuery user interface; and (c) web service with the REpresentational State Transfer (REST) API. The latter two were implemented using the Web Service Gateway Interface (WSGI) running under Apache HTTP Debian GNU/Linux server with the mod_python server module. The application inputs are: the AL component, one or more tests (optional), parameters values, initial conditions and inputs to the analogue and event ports. The application produces the model report (in PDF

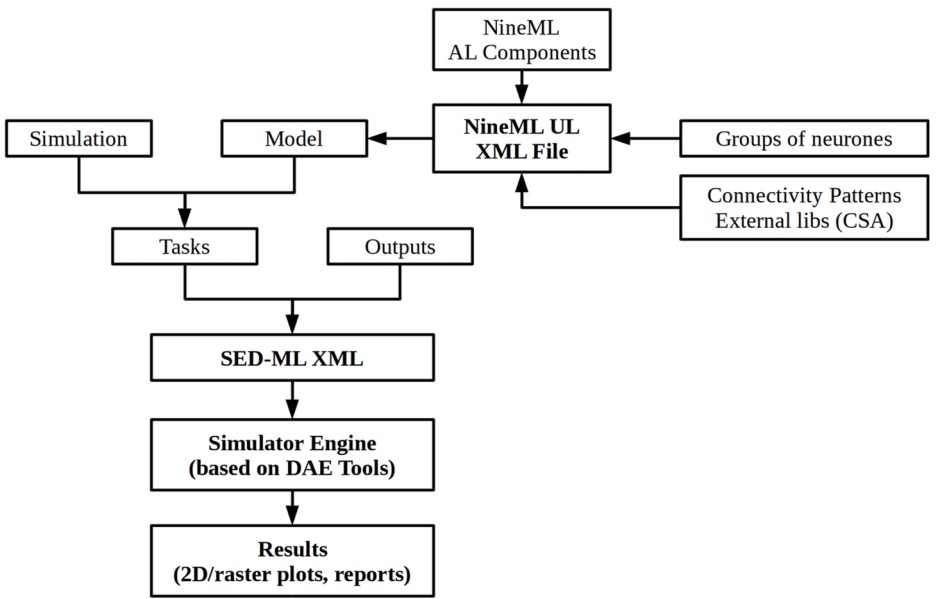

**Figure 12** An overview of the NineML reference implementation simulator.

or html format) and tests results (variable plots). The role of DAE Tools software is to process NineML xml input files, generate the model structure using the lib9ml library, execute the server-side simulations, produce reports and deliver them to the clients.

### NineML reference implementation simulator

The NineML reference implementation simulator represents a small scale simulator for testing and validating purposes, with the full support for the NineML language. All input information are given in a simulator-independent format: NineML xml files (mathematical models) and SED-ML xml file (simulation settings). SED-ML is the Simulation Experiment Description Markup Language an xml-based format for encoding simulation experiments (http://sed-ml.org). It is based on the Minimal Information About a Simulation Experiment guidelines (MIASE, http://biomodels.net/miase) and used to specify: (a) the definition of the model(s) to be used; (b) the definition of the simulation settings to be used; (c) the experimental tasks to be run; and (d) the post processing of the results and outputs (2D-plots, 3D-plots, data tables etc.). The simulator utilises the fundamental modelling concepts in DAE Tools: parameters, variables, equations, ports, models, state transition networks and discrete events as a basis for implementation of the higher-level concepts from the NineML language such as neurons, synapses, connectivity patterns, populations of neurons and projections. Again, the role of DAE Tools software is to process NineML and SED-ML xml input files, generate the model structure, execute the simulation, and produce the results based on inputs from SED-ML file. The simulator implements the synchronous (clock-driven) simulation algorithm and the system of equations is integrated continuously using the variable-step variable-order backward differentiation formula using Sundials IDA DAE solver. The exact event times (spike occurrences) are calculated by detecting discontinuities in model equations using root functions. An overview of the simulator is presented in Fig. 12.

## CONCLUSIONS

DAE Tools modelling, simulation and optimisation software, its programming paradigms, the main features and capabilities have been presented in this work. Some shortcomings of the current approaches to mathematical modelling have been recognised and analysed, and a new hybrid approach proposed. The hybrid approach offers some of the key advantages of modelling languages paired with the flexibility of the general purpose languages. Its benefits have been discussed such as the support for the runtime model generation, runtime simulation set-up and complex runtime operating procedures, interoperability with the third party software packages, and embedding and code-generation capabilities. The software architecture and the procedure for transformation of the model hierarchy into a DAE system as well as the algorithm for the solution of the DAE system have been presented. The most important modelling concepts available in the DAE Tools API required for model development and simulation execution have been outlined.

The software has successfully been applied to two different scientific problems. In the first example, the authors took advantage of the object-oriented characteristics of the software and the interoperability with the NumPy library for the development of a model hierarchy to mathematically describe operation of lithium-ion batteries at different physical scales. In the second example, the DAE Tools software has been used as a reference implementation simulator for the new XML-based domain specific language (NineML). DAE Tools embedding capabilities have been utilised to provide a simulator available in three versions: (a) desktop application, (b) web application and (c) web service.

The current work concentrates on a further support for systems with distributed parameters (i.e. high-resolution finite volume schemes with flux limiters), the additional optimisation algorithms and the parallel computation using the general purpose graphics processing units and systems with the distributed memory. The parallel computation will rely on the code generation capabilities to produce the C source code for the DAE/ODE solvers that support the MPI interface such as PETSc and Sundials IDAS/PVODE, including the data partitioning and the routines for the inter-process communication of data.

## ACKNOWLEDGEMENTS

The case study *Multi-scale model of phase-separating battery electrodes* has been performed and the corresponding subsection written by Raymond B. Smith, Department of Chemical Engineering, Massachusetts Institute of Technology, Cambridge, Massachusetts, USA.

### Funding

The author received no funding for this work.

### Competing Interests

Dragan Nikolić is an employee of DAE Tools Project, Belgrade, Serbia.

## Author Contributions

- Dragan D. Nikolić analyzed the data, wrote the paper, prepared figures and/or tables, performed the computation work, reviewed drafts of the paper.

## Data Deposition

http://www.daetools.com.

http://sourceforge.net/projects/daetools.

## Supplemental Information

Supplemental information for this article can be found online at http://dx.doi.org/10.7717/peerj-cs.54#supplemental-information.

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
