# Peer review of "DAE Tools: equation-based object-oriented modelling, simulation and optimisation software"

_PeerJ Computer Science, doi:10.7717/peerj-cs.54_

## Round 0.1 · original submission · Major Revisions

· Academic Editor

Major Revisions

Three reviewers have evaluated your work - please address all their comments.

Staff note: One reviewer noted that this article was previously published as a poster at PSE2015 / ESCAPE 25. This fact should be acknowledged / noted in this article.

·

Basic reporting

In this work, the author presents DAE Tools, a software for building, simulating, and optimizing models comprised of DAEs. This software takes an approach in which the author combines the versatility of general-purpose programming with an interface resembling that of a modeling language. The software appears mature, and includes significant online documentation and a tutorial. These are all welcome, and contribute to my overall impression that this is a well-developed project suitable for publication in PeerJ Computer Science. However, minor revisions must first be made.

The Introduction provides a detailed description of the state of the field, and shows the relevance of DAE Tools. However, it is also relatively lengthy and verbose, and could be condensed without significant loss of descriptiveness. Lines 192-230 in Features read similarly to lines 20-47 and 59-71 of the Introduction. To address this, I propose condensing lines 31-71 to outline the limitations of existing modeling frameworks and the specific goals of DAE Tools: combining the qualities of domain-specific and general-purpose languages to handle model specification, simulation setup, execution of a simulation and/or optimization, and processing the results. The Features section should (and does) expound on how DAE Tools addresses these shortcomings.

The Architecture and Solution of a DAE System sections are well-constructed. While useful for inclusion in the Documentation (or a supplement), I do not think the class diagrams Figs. 1-7 are necessary for the main text of the manuscript. I did find Figs. 8-11 (UML sequence diagrams) to be useful.

The tone is generally refined, however there are several cases of informal phrases (e.g. "sort of").

The future aims of the project are briefly mentioned in the Main Features section, or alluded to ("only Order-1 DAEs, at the moment")​. Are there significant plans for further expansion of this software? If so, it may be worth describing these plans.

Typographical changes:
- "log" is listed twice, line 337 and 379
- The definite article "an" should be used before "API." ("an API")
- C++ and Python should be capitalized throughout the manuscript

Experimental design

In this portion of the review, I have described minor bugs in the software or documentation. None of these changed drastically affect the conclusions of the work.

- SourceForge currently suggests 1.3.0 as the most recent release, yet 1.4.0 is described on the website. Since installation is significantly different for the versions, it would be useful to update SourceForge on the most recent version.

- Documentation describing installation of DAE Tools lists dependencies, but should also list versions (e.g. NumPy >= 1.6.1).

- The Docs (5.2.1) do not include import statements at the top. While it can be inferred, please include ``from daetools.pyDAE import *''

- The Docs (5.2.1 & 5.3.1) do not work in Python as-is. Specifically, self.m = myModel("myModel", "Description") in 5.3.1 is input to the myModel object as myModel(name="myModel", parent="Description"), thus the Parent is rejected.

- Please rename the Listings files (if allowed by the Journal) to replaces spaces with underscores to allow simple execution in Python.

- When running Listings 1 & 2 from the command line, they work well. However, executing these files line by line in a Jupyter console throws an error due to the long printout--is there a way to silence it?

- The listings of code in the manuscript should match the listings of code in the supplementary material, for consistency.

Validity of the findings

As the underlying solvers used in this toolkit are well-established, there is no question of their validity.

The examples used in the manuscript have also been published independently of the software.

Comments for the author

Although not essential for publication, I recommend the author to consider using PEP8 (https://www.python.org/dev/peps/pep-0008/) conventions (especially naming styles) in the future for consistency with other Python packages.

Similarly, I would find it useful for this software to be available on GitHub so that users can easily compare versions, submit bugs, etc.

Reviewer 2 ·

Basic reporting

The manuscript describes DAE Tools, a software tool for equation-based object-oriented modelling, simulation, and optimization of systems on differential-algebraic form. DAE Tools combines the strength of DAE modelling languages such as, e.g., Modelica, with the expressiveness and generality of ordinary programming languages such as Python and C++ by providing a modelling language API that can be used from Python or C++. After the introduction the paper focuses on the DAE Tools architecture and the operation of that are performed and the objects involved when a simulation is performed.Then a small example (CopperBlock) is used to illustrate the steps involved in modelling and simulation of DAE Tools. Finally some other DAE Tools applications are briefly described.

DAE modeling, simulation and optimization tools is an area that is currently attracting a lot of interest. Strong points of DAE Tools is 1) the tight integration with Python/C++ and 2) the fact that it is open source. However, there are other such tolls available. JModelica.org is a open source Modelica tool that is very well integrated with Python. In spite of this I cannot find a single reference to, or comparison with, this. Another example is "AP Monitor" which also supports modeling, simulation, and optimization of DAE systems. Still no references or comparisons.

DAE Tools also appears to lack vital functionality for DAEs. For example it currently only support index-1 DAE systems. Competing systems such as JModelica.org supports high-index DAEs through index reduction using Panetlides algorithm and dummy derivatives. Also, it does not seem as if DAE Tools currently supports discrete-time sampled systems. This is problematic since a major use case for DAE tools is simulation of closed control loops where the controller are implemented in discrete-time.

The manuscript is dominated by large figures containing UML class diagrams and UML sequence diagrams. I doubt that this is really useful for the readers. The same holds for the code examples. They are nice, but why duplicate them in both Python and C++. For a scientific paper like this would be enough to show the examples in one language, e.g., Python and then simply say that one alternatively could use C++ to express the same thing. It would have better if this page space had been spent on better describing the functionality provided by the modeling language API.

The final "Applications" section is too short to really explain the applications

Experimental design

No Comments

Validity of the findings

No Comments

Comments for the author

Please focus your paper on 1) what is unique with DAE Tools from a DAE modeling point of view and 2) compare with alternative tools such as JModelica.org and AP Monitor. Focus less on the architecture.

Reviewer 3 ·

Basic reporting

This manuscript describes the architecture, design philosophy, and structure of DAE Tools, a package for modeling and solving simulation and optimization problems. DAE Tools combines strengths from modeling languages and general-purpose programming languages to allow runtime model generation and simulation set-up, complex runtime operating procedures (for hybrid discontinuous systems), model exchange and co-simulation, etc.

The project itself seems quite interesting and the manuscript is mostly well written and organized. As such I recommend the submission to be accepted, pending some minor revisions, as listed below.

See "General Comments for the Author"

Experimental design

See "General Comments for the Author"

Validity of the findings

See "General Comments for the Author"

Comments for the author

Main comments/questions

1. My main concerns relate to the possible loss of flexibility in modeling and simulation when compared, for example, with user-defined models simulated with libraries such as PETSc or Sundials. The tools provided by such libraries provide extensive user control which does not seem to have been completely exposed in DAE Tools. Using the IDAS solver in Sundials as an example (since this appear to be the workhorse in DAE Tools), I hope that the author can provide clarifications on the following comments:

1.1 From the UML diagram in Fig. 10, it seems that the only user-defined functions relate to the definition of problem parameters, variables, and equations. However, IDAS itself provides more flexibility, e.g. specifying Jacobian-related information. Does DAE Tools expose this capability? Or does it always rely on AD-generated Jacobians? There are situations when an exact Jacobian may be overkill and a user-supplied Jacobian approximation (especially for large, multi-scale problems) leads to much more efficient solutions.

1.2 The statement on line 186 seems to suggest that DAE Tools also exposes the sensitivity analysis capabilities of IDAS. However, no further mention is made of this. If calculating sensitivities is supported, how are the sensitivity equations generated? IS it also AD-based? Also, what are the limitations in terms of problems that can be addressed (given that IDAS cannot currently perform sensitivity analysis of hybrid discontinuous systems)?

2. There is no mention of support for parallel computation (either distributed or shared-memory), a main area of applications for both PETSc and Sundials, which were designed with large-scale problems in mind. Are there any plans on developing DAE Tools along those lines?

3. The accuracy and efficiency comparisons between DAE Tools and Matlab's ode15s (lines 728 - 736) are questionable. First, it is mentioned that the two integrators were using "default solver tolerances for both" and that the "simulation outputs were indistinguishable". Since both ode15s and IDAS allow user specified tolerances, why not set them equal to each other and then perform a more quantitative comparison of the achieved accuracy? Furthermore, there are many other possible reasons for efficiency differences between the two beyond those listed in the manuscript (algorithmic differences and language differences being two of them)

Minor comments

1. The very first classification of mathematical modeling (lines 10-19) differentiates between modeling languages (Modelica, gPROMS, etc.) on one hand and what I view more as solver libraries (PETSc and Sundials) on the other hand. Indeed, neither PETSc nor Sundials really offer proper modeling support (a view that seems to also be acknowledged by the author later on, see lines 70-71). In this case, is this classification really appropriate?

2. Is the author aware of the Assimulo project? If so, what (if anything) is common between these two efforts of providing simpler interfaces to solvers such as those in Sundials?

3. The term "degrees of freedom" (lines 242-244) is used here in a rather unconventional sense. While one can better infer what their meaning is herein (from the description of the CopperBlock simulation), it may be useful to clarify that early on, when the term is first introduced (or else maybe come up with a better name).

4. Since DAE Tools relies on a "heavily modified ADOL-C", what (if any) are the plans on staying up-to-date and incorporate possible new releases of ADOL-C?

Language, grammar, manuscript structure and organization

1. While the manuscript reads pretty well, there are places that sound as if they were written by someone else! The Abstract and Conclusions sections, as well as the last paragraph in the Introduction, stand out in this respect. In particular: using past tense in the Abstract and at the end of the Introduction section (before anything has really been presented); very repetitive statements made in the Conclusions section.

2. There are numerous missing or incorrectly used definite and indefinite articles throughout the manuscript.

3. The description of the various interfaces in the six DAE Tools packages (in the Architecture section) is very difficult to parse (the one for the "core" package is effectively a 20-line long sentence). These may be better presented in tables or else bulleted lists.

4. "linear" is misspelled on line 304

---

## Round 0.2 · accepted · Accept

· Academic Editor

Accept

I am pleased that PeerJ will be publishing this article.